# Detection of Antioxidant Phytochemicals Isolated from *Camellia japonica* Seeds Using HPLC and EPR Imaging

**DOI:** 10.3390/antiox9060493

**Published:** 2020-06-05

**Authors:** Chalermpong Saenjum, Thanawat Pattananandecha, Kouichi Nakagawa

**Affiliations:** 1Cluster of Excellence on Biodiversity Based Economics and Society (B.BES-CMU), Chiang Mai University, Chiang Mai 50200, Thailand; thanawat.pdecha@gmail.com; 2Department of Pharmaceutical Sciences, Faculty of Pharmacy, Chiang Mai University, Chiang Mai 50200, Thailand; 3Division of Regional Innovation, Graduate School of Health Sciences, Hirosaki University, 66-1 Hon-cho, Hirosaki 036-8564, Japan

**Keywords:** *Camellia japonica* seed, antioxidants, EPR imaging, HPLC, free radical

## Abstract

In this study, we investigated the formation of stable radicals and compounds related to antioxidants in *Camellia japonica* seeds using high performance liquid chromatography (HPLC) and X-band electron paramagnetic resonance imaging (EPRI). The *C. japonica* seed coat extracts exhibited antioxidant activity in both in vitro and cell-based studies. The extracts inhibited reactive oxygen and reactive nitrogen species production in cell-based studies. HPLC chromatograms indicated that hydrophilic antioxidant compounds—namely, gallic acid, gallocatechin, epigallocatechin, caffeine, catechin, epicatechin, and epicatechin gallate—were found in the methanolic extract. Lipophilic antioxidant compounds—including α-tocopherol, γ-tocopherol, β-tocopherol, δ-tocopherol, α-tocotrienol, γ-tocotrienol, and δ-tocotrienol—were found in the hexane extract. EPRI primarily detected paramagnetic species in seed coats. These radical species were stable organic radicals based on the peak-to-peak line width and *g*-values. The signals from these stable radicals were strong and stable with a *g*-value of 2.002. Noninvasive EPRI of the radicals present in *C. japonica* seeds indicated that the stable radicals were essentially located in the seed coats. The EPRI of the cotyledon demonstrated that additional radicals were localized at an apex of the cotyledon. The results indicated that the stable radicals detected by EPRI and the hydrophilic and lipophilic antioxidant compounds analyzed by HPLC were related to antioxidant reactants and products.

## 1. Introduction

The tree *Camellia japonica* (*C. japonica*) flowers between January and March. The flowers usually last three to four weeks, and, during this time, the fruit develops. Each cavity of the spherical fruit contains up to two small brown seeds that are no more than two centimeters in diameter. The cotyledons of these seeds are used in food as well as cosmeceuticals. In addition to minerals, such as calcium and magnesium, the seeds also contain fatty acids, namely, oleic acid and stearic acid. Oleic acid, which is a monosaturated fatty acid, has been reported to have health benefits in humans [1,2]. However, many other features of *C. japonica* seed coats remain to be determined.

In recent years, reactive oxygen species (ROS) have been considered to be involved in seed preservation and other physiological phenomena [3,4]. ROS are produced continuously during seed development and seed storage. They can react with various phytochemical antioxidants found in plant seeds, such as anthocyanins, polyphenols, and flavonoids, for the production of stable organic radicals in plant seeds. The detection of free radicals using noninvasive techniques can provide detailed information regarding how ROS-related free radicals are involved in plant physiology. Dark colored seeds may contain antioxidants. In the majority of cases, the color of the seed is related to the antioxidant, such as anthocyanin and anthocyanidin-related compounds [5,6]. The antioxidant processes produce reaction products, which may stable intermediate radicals of antioxidants. The stability (or reactivity) of the intermediate radical of the antioxidant compound is also an important feature of antioxidants. If the intermediate radical is stable, the intermediate may not react further in the system. The information on stable radicals in the seeds is very limited; reports previously demonstrated that the antioxidant determination in plant seeds required the preparation of the extract using organic solvents, which produce organic waste [5,6,7]. The current study proposed an antioxidant determination technique in plant seeds using a non-invasive technique called electron paramagnetic resonance.

Electron paramagnetic resonance (EPR), also known as electron spin resonance (ESR) spectroscopy, exploits the electron-spin resonance phenomenon. This also measures the resonant microwave power absorption spectra of unpaired electrons manifested in a constant magnetic field in an atom, a molecule, or a compound. Limited research has been done on endogenous paramagnetic species. EPR imaging is a powerful, non-invasive, and non-destructive technique for measuring the spatial distribution of paramagnetic species, including stable organic radicals, transition metal complexes, and transition metal ions. Additionally, EPR is a sensitive and non-destructive method for detecting paramagnetic species or free radicals in plant samples at ambient temperatures [8,9]. This method has been used in several studies to investigate naturally occurring free radicals in plant samples [8,9,10,11,12]. EPR also provides insights into how radicals are spatially distributed. Detailed studies on EPR spectra, such as line intensities and line widths between different radicals, provide information on both radical generation processes and radical moieties [10,11]. The information regarding the spatial distribution of radicals is a useful indexing tool for assessing antioxidant activity over time. Therefore, the EPR method could be useful for identifying the endogenous radicals related to the scavenging activity in *C. japonica* seeds; however, the pigments (naturally coloring the plant tissue) in the seed coat have not been investigated to date.

In the present study, two types of solvent (methanol and hexane)-extracted compounds from *C. japonica* seed coats were examined by high performance liquid chromatography (HPLC) to analyze for hydrophilic and lipophilic phytochemical antioxidant compounds. Stable radicals related to antioxidants in untreated *C. japonica* seeds and seed coats were investigated using X-band EPR and electron paramagnetic resonance imaging (EPRI). Additionally, their antioxidant abilities and related reactions in both in vitro and cell-based studies were also investigated.

## 2. Materials and Methods

### 2.1. Chemicals and Reagents

Catechin and related compounds were obtained from Sigma Chemical Co., Ltd., (St. Louis, MO, USA). δ-, β-, γ-, and α-tocotrienol and δ-, β-, γ-, and α-tocopherol were purchased from Merck Co., Ltd., (Kenilworth, NJ, USA). All the chemicals and solvents used were either standard-, analytical-, or HPLC-grade and were purchased commercially from Fluka Chemical Co., Ltd., (Buchs, Switzerland), Sigma Chemical Co, Ltd., and Merck Co., Ltd., All the chemicals and reagents used in the cell-based research were purchased from Invitrogen (Waltham, MA, USA) and Roche (Mannheim, Germany).

### 2.2. Camellia japonica Seeds Sample

Experiments were carried out on *C. japonica* seeds harvested from an organic farm located in Izu-Ohoshima, Tokyo, Japan at the beginning of spring 2016. The seeds were provided by Ohoshima Tsubaki Co., Ltd., (Tokyo, Japan) as a gift. The *C. japonica* seeds were stored at room temperature (RT, 20 °C) in a closed, plastic bottle. The seeds were divided into seed coats and cotyledons, and dried for 24 h at RT before EPR measurements. All the samples were used without subjecting them to any pre-treatments. In addition, the spin concentration of the EPR signal was calculated as previously described [8].

### 2.3. Preparation of Extracts

The procedure of sample extraction was slightly modified from Watanabe et al. (2014) [13]. Briefly, the seed coats of *C. japonica* were cleaned, dried, and ground in iceboxes into coarse powder. The powder was separately extracted with 80% methanol at 70 °C and hexane at 60 °C using a shaking incubator at 120 rpm for 2 h. The solutions were collected, and the solvent was evaporated under reduced pressure then vacuum dried to obtain methanolic extracts (ME) and hexane extracts (HE).

### 2.4. Determination for In Vitro Antioxidant Activity

#### 2.4.1. Scavenging Effects on Nitric Oxide (NO)

The scavenging effects on NO were investigated following the method of Sreejayan and Rao (1997) [14] and the improved method of Kidarn et al. (2018) [15]. Briefly, 200 µL of various concentrations of tested samples or the positive control, curcumin, were added to 800 µL of 6.25 mM sodium nitroprusside in phosphate buffer saline (PBS) at pH 7.4. The reaction mixture was incubated at 37 °C for 150 min. After the reaction time, we transferred 150 µL of the mixture solutions into a 96-well plate, and 100 µL of Griess reagent—prepared by mixing of 0.1% (*w*/*v*) naphthylethylenediamine dihydrochloride with 1% (*w*/*v*) sulfanilamide in 5% phosphoric acid at a ratio of 1:1—was added to form the color reaction and incubated for 5 min. Then, the absorbance was measured at 540 nm by spectrophotometry and the results were expressed as 50% inhibition concentration (IC_50_).

#### 2.4.2. Scavenging Effects on the Superoxide Anion

The effects of scavenging activity on the superoxide anion radical was determined using the method of Yangping et al. (2004) [16]. Initially, the reaction mixture consisted of 78 µM β-nicotinamide adenine dinucleotide, 25 µM nitroblue tetrazolium, 45 µM ethylenediaminetetraacetic acid, and various concentrations of the tested sample in a final concentration range of 10–200 µg/mL in PBS at pH 7.4. The mixtures were then added to 25 µL of 20 µM phenazine methosulphate to start the reaction and then stored at room temperature for 5 min with light protection. Finally, we immediately measured the absorbance of the completed reaction using a multimode detector at the wavelength of 560 nm. L-ascorbic acid and catechin were used as positive controls. All the samples were done in triplicate, and the results were expressed as IC_50_.

#### 2.4.3. Inhibitory Effect on Lipid Peroxidation

The inhibition effect on linoleic acid peroxidation was evaluated using the modified method developed by Saenjum et al. (2012) [17]. Briefly, the reaction mixture consisted of 20 mM linoleic acid emulsion, 20 mM L-ascorbic acid, 100 mM Tris-HCl at pH 7.5, and various concentrations of the tested samples and positive controls (catechin and α-tocopherol). The reaction mixture was initiated by the addition of 40 mM Fe_2_SO_4_·7H_2_O and then incubated at 37 °C for 30 min. Then, 40% (*v*/*v*) trichloroacetic acid and 1% (*w*/*v*) thiobarbituric acid in 50 mM NaOH were added to stop the reaction and heated at 100 °C for 10 min to initiate the color reaction. Finally, all of the mixture solutions were spectrometrically measured at 532 nm. The percentage of linoleic acid peroxidation inhibition was calculated. All the samples were done in triplicate, and the results were expressed as IC_50_.

### 2.5. Determination of Antioxidant Activity in Cell-Based Study

#### 2.5.1. Determination of Inhibitory Effect on the Intracellular ROS Production

The inhibitory effect of the extracts on the intracellular ROS production stimulated by hydrogen peroxide (H_2_O_2_) was determined using the slightly modified dichloro-dihydro-fluorescein diacetate (DCFH-DA) method of Banjerdpongchai et al. (2016) [18]. Peripheral blood mononuclear cells (PBMCs, 1 × 10^6^ cells/mL in a 96-well culture plate) were pre-treated with 0–200 µg/mL of tested samples for 6 h, followed by treatment with 500 µM H_2_O_2_ for 30 min to initiate intracellular ROS production. Then, the cultured cells were removed and given a fresh medium followed by incubation at 37 °C and 5% CO_2_ for 12 h. Finally, we added 40 µM DCFH-DA solution to each mixture and incubated at 37 °C with 5% CO_2_ for 30 min. The green fluorescent intensity was measured the excitation and emission wavelengths using a fluorescent microplate reader at 480 and 525 nm, respectively. N-acetyl cysteine (NAC), catechin, and L-ascorbic acid were used as positive controls.

#### 2.5.2. Determination of NO and Inducible Nitric Oxide Synthase (iNOS) Production

The inhibitory effect on NO and iNOS production induced by combined lipopolysaccharide (LPS) and interferon-γ (IFN-γ) was determined using the method of Hong et al. (2002) [19] and Hu et al. (2003) [20]. Briefly, the murine macrophage cells (RAW 264.7) were cultured in Dulbecco’s modified Eagle’s medium (DMEM) supplemented with 10% fetal bovine serum, 100 units/mL of penicillin, and 100 µg/mL of streptomycin. RAW 264.7 cells were pre-incubated in 24-well plates at 37 °C with 5% CO_2_ for 12 h. Then, the cultured cells were removed and given a fresh medium containing different concentrations of the tested samples in final concentrations ranging from 10 to 100 µg/mL. After incubation for 12 h, we added LPS and IFN-γ, to final concentrations of 2 ng/mL and 50 pg/mL, respectively, and then continued incubation at 37 °C with 5% CO_2_ for 72 h. The supernatants of the cultured medium were collected to measure the NO by measuring the color formed with the Griess reagent at 540 nm. We calculated the NO using a calibration curve of potassium nitrite, and fresh cultured medium was used as a blank. The cells were lysed using CelLytic^TM^ M cell lysis buffer to obtain the cell lysate for iNOS measurement. A mouse iNOS ELISA kit (CSB-E08326M, Cusabio Biotech, Co., Ltd., Houstan, TX, USA) was used for measuring the level of iNOS. Curcumin and catechin, naturally anti-inflammatory and antioxidant compounds, were used as positive controls. The cellular DNA was quantified using a Quant-iT PicoGreen Assay (Invitrogen, P11496) according to the manufacturer’s protocols. The total protein concentration in the cell lysates was measured by the Bradford protein assay. The cell viability of the control samples and those stimulated with LPS and IFN-γ for 72 h was determined using a cell proliferation reagent, WST-1 [21].

### 2.6. Chromatographic Analysis of Polar Phenolic Compounds

Polar phenolic compounds, namely catechin, epicatechin, epicatechingallate, epigallocatechin, epigallocatechingallate, gallocatechin, gallocatechingallate, gallic acid, and caffeine, were analyzed by reverse-phase HPLC using an Agilent 1200 equipped with a multi-wavelength detector. The assay was carried out using a Symmetry Shield RP18 column (4.6 mm × 250 mm, 5 µm particle diameters, Waters Co., Ltd., Milford, MA, USA), and 10% acetonitrile in 0.1% acetic acid and de-ionized water were used for the mobile phase at a flow rate of 1.0 mL/min with the detection wavelength at 280 nm.

### 2.7. Chromatographic Analysis of Tocotrienols and Tocopherols

The δ-, β-, γ-, and α-tocotrienol and δ-, β-, γ-, and α-tocopherol contents of the ME and HE extracts were analyzed by reverse-phase HPLC [22] using an Agilent 1200 with excitation and emission wavelengths of the fluorescence detector at 296 nm and 330 nm, respectively. The assay was carried out using a KINETEX^®^ PFP column (150 mm × 4.6 mm, Phenomenex Co, Ltd., Torrance, CA, USA), and a mixture of methanol and de-ionized water at a 9:1 ratio was used for the mobile phase at a flow rate of 0.6 mL/min.

### 2.8. EPR

#### 2.8.1. EPR Measurements

A JEOL RE-3X 9 GHz EPR spectrometer (JEOL Resonance Inc., Tokyo, Japan) was selected to measure the continuous wave (CW). The operation system of the X-band mode was programmed at 9.44 GHz and a 100 kHz modulation frequency. All of the CW EPR spectra were obtained with a single scan. The following typical CW EPR settings were used: microwave power, 5 mW; time constant, 0.1 s; sweep time, 4 min; magnetic field modulation, 0.3 milli Tesla (mT); and sweep width, 10 mT. All of the measurements were performed at ambient temperature. The *C. japonica* seeds were sequentially inserted into an EPR tube and/or were attached to an EPR rod for each measurement. The EPR signal intensity was divided by the sample weight in order to normalize each intensity.

#### 2.8.2. EPR Imaging Measurements

A JEOL RE-3X 9 GHz EPR spectrometer was modified for use as an EPR imager by attaching magnetic field gradient coils (Yonezawa Densen Ltd., Yonezawa, Japan) and their power supplies. The X- and Y-axes of the gradient coils in anti-Helmholtz coil configurations were used for 9 GHz EPR imaging. The gradient coils were cooled with a water thermostat at 16 °C to avoid overheating. Takasago BWS 60-5 bipolar power supplies (Takasago Ltd., Tokyo, Japan) were used. The maximum available field gradient along the X- and Y-axes was 3.4 mT/cm. The magnetic field of the EPR spectrometer was current-stabilized to perform EPR imaging. The field-based control is incompatible with the application of the magnetic field gradients. The instrument and the gradients were controlled using the SpecMan4EPR (FeMi Instruments LLC, Chicago, IL, USA) software. All measurements were performed at ambient temperature.

Sixteen equal-angle spaced projections obtained with a maximum gradient of ~3.4 mT/cm were used. The first-derivative EPR spectra were numerically integrated to obtain the corresponding absorption spectra. The 2D images were reconstructed from a complete set of projections, which were collected as a function of the magnetic field gradient. Before reconstruction, each projection was deconvolved using fast Fourier transformation with the measured zero-gradient spectrum to improve the image resolution. The 2D image reconstruction was performed using the back-projection algorithm in the EPR-IT software package from the Center for EPR Imaging in Vivo Physiology at the University of Chicago [23]. All the measurements were performed at ambient temperature. A detailed description is available elsewhere [11].

### 2.9. Statistical Analysis

The obtained results are expressed as the mean ± standard deviation of three independent experiments. The statistical analysis was performed using one-way analysis of variance. Significant differences at the level of *p* < 0.05 were determined by Tukey’s multiple comparison test.

## 3. Results and Discussion

### 3.1. Antioxidant Activity

Hydrophilic (ME) and lipophilic (HE) *C. japonica* seed coat extracts were determined for antioxidant activity in both ROS and RNS systems using in vitro and cell-based studies. The results from in vitro antioxidant activity measurements, including nitric oxide scavenging (RNS), superoxide anion scavenging (ROS), and the inhibition of lipid peroxidation, are shown in Table 1. The ME of *C. japonica* seed coats exhibited a greater hydrophilic antioxidant (nitric oxide and superoxide anion scavenging) than that of the HE of *C. japonica* seed coats. The HE of *C. japonica* seed coats exhibited a higher antioxidant activity through an inhibitory effect on lipid peroxidation than that of the ME of *C. japonica* seed coats. The results indicated that *C. japonica* seed coats contain both hydrophilic and lipophilic antioxidants, which corresponds to the studies reported previously by Lee and Yen (2006) [24] and Chiyana et al. (2018) [25], who reported that the methanolic extract of tea seed oil from *C. oleifera* Abel. showed strong antioxidant activity as investigated by 1,1-diphenyl-2-picrylhydrazy radical (DPPH) scavenging activity and that *C. assamica* seed oil exhibited potent antioxidant activity through DPPH scavenging activity and inhibition of lipid peroxidation by the ferric thiocyanate method. ME and HE did not affect the total DNA, protein production, or cell viability. The viability of RAW 264.7 cells upon exposure to both ME and HE at a concentration range of 10–100 ppm remained the same as that of the control. The amount of the DNA, which indicates the ability of the cells to proliferate, clearly showed that ME and HE did not activate cell proliferation. Both ME and HE did not alter the ability of the cells to produce total proteins. The inhibitory effects of ME and HE on nitric oxide and iNOS production, in combination with LPS-IFN-γ, on RAW 264.7 cells are shown in Table 2, and the inhibitory effect on intracellular ROS production in PBMC cells is illustrated in Figure 1. The PBMC cells were pretreated with ME, HE, NAC, catechin, and L-ascorbic acid for 6 h and then treated with 500 µM H_2_O_2_ for 30 min. After 30 min of incubation, the ROS scavenging activity was approximately 35% and 48% for 100 and 200 µg/mL of ME, respectively, and approximately 25% for 200 µg/mL of HE. The results indicated that ME and HE protected against H_2_O_2_-induced intracellular ROS production. Both the methanolic and hexane extracts exhibited antioxidant activity through an inhibitory effect on ROS and RNS production in the cell-based study. The ME of the *C. japonica* seed coat extract exhibited a greater antioxidant activity in the cell-based study than that of HE of the *C. japonica* seed coat.

### 3.2. Chromatographic Analysis of Biochemical Compounds

Figure 2 displays the HPLC chromatogram identifying catechin and its derivatives. The ME of *C. japonica* seed coats contained hydrophilic antioxidant compounds, namely gallic acid, gallocatechin, epigallocatechin, caffeine, catechin, epicatechin, epigallocatechin gallate, and epicatechin gallate, which were not detected in the HE of *C. japonica* seed coats. The largest peak of ME corresponded to catechin. Additionally, Figure 3 displays the chromatograms of the hexane-extracted compounds and control standards. The chromatogram indicates that tocotrienols and tocopherols were extracted by hexane from the seed coat of *C. japonica*, which was composed of α-tocopherol, γ-tocopherol, β-tocopherol, δ-tocopherol, α-tocotrienol, γ-tocotrienol, and δ-tocotrienol. The results were correlated to a previous report by Hu and Yang (2018) [26], who reported that the major lipophilic antioxidant phytochemicals in *C. oleifera* oil were α-tocopherol and γ-tocopherol. The amounts of phytochemical compounds in the ME and HE extracts is shown in Table 3. The current results indicated that the catechin and related compounds (hydrophilic antioxidants) in the ME of *C. japonica* seed coats possessed both ROS- and RNS-scavenging activities. These compounds also suppressed the production of ROS and RNS in the cell-based study of *C. japonica* seed coats, as represented by both the hydrophilic and lipophilic antioxidant phytochemicals. Catechin and the related compounds all have phenolic hydroxyl groups that were able to stabilize the free radicals [23]. The inhibition on RNS production in RAW 264.7 cells and the scavenging effect on intracellular ROS production by *C. japonica* seed coat extracts corresponded to antioxidant phytochemical compounds. Catechin and related compounds all have phenolic hydroxyl groups that are able to stabilize the free radicals [27]. The phenolic hydroxyl groups of catechin and its derivatives can react with ROS and RNS in a termination reaction, resulting in a breakdown of the cycle of new radical production [28]. Furthermore, the tocotrienols and tocopherols present in the HE of *C. japonica* seed coats included a lipophilic antioxidant. The HE exhibited a higher antioxidant activity in the cell-based study than in the in vitro study. Vitamin E isoforms (tocopherols and tocotrienols) were incorporated into cellular membranes and exhibited intracellular antioxidant effects. Additionally, tocopherols and tocotrienols exhibited similar mobilities within the membranes; however, tocotrienols were more easily transferred through the membranes and incorporated into the membranes [29,30]. Therefore, we hypothesized the antioxidant activity from the combination of hydrophilic and lipophilic antioxidant phytochemicals of *C. japonica* seeds.

### 3.3. EPR of Stable Radicals in C. japonica Seed Coat

Figure 4 displays the wide-range EPR spectrum in order to determine any paramagnetic species of whole *C. japonica* seeds at ambient temperature. The magnetic field center used was 240.0 mT with a 300 mT sweep width. The EPR spectrum had one distinguishable signal with a broad line. The signal was stable, and the EPR results were similar for at least a month. No Mn^2+^ signal was observed in the seed coats, as previously described in other studies [8,10,11]. The signal was strong and reproducible with a *g* value of about 2.002, which is indicative of stable organic radicals [6,8]. The *g* value gives a clue as to the state of the unpaired electron in the compound.

In addition, antioxidants such as phenolic compounds that lose the hydrogen of the OH-group become intermediate radicals. The unpaired electron (radical) is delocalized throughout the 2p_z_ orbitals of the conjugated double bond. The delocalization of the unpaired electron (radical intermediate) is stable and otherwise not reactive. This suggests that these radicals can be generated under oxidative conditions and that antioxidant-related organic compounds were present in the seed at ambient temperature.

Stable radicals detected by EPR can be the products of antioxidant reactions with ROS and/or the oxidation of compounds in the seed coats. There is a possible explanation for stable radical production. We propose that a stable (or reactive) intermediate may be responsible for antioxidant reactions and the scavenging activity in the seed coats, as suggested previously [31]. In most cases, unpaired electrons or radicals can be delocalized within a molecule for stability. In addition, plant physiological processes produce ROS [31,32], which, together with nitric oxide, regulate various processes in plants [29].

Figure 5 displays the EPR spectra obtained at g values of ~2.002 of the *C. japonica* seed coats at ambient temperature. Additionally, the spin (radical) concentration was estimated using a TEMPOL (known concentration) solution in a capillary tube (outer diameter 1.0 mm, inner diameter 0.9 mm). The number of spins per gram was approximately 3 × 10^18^. The calculation procedure was performed as described previously [11,12]. The radical species between the seed coats and cotyledon are not the same species, as each composition is not the same. The radical intensity of the seed coats is approximately 22 times stronger than that of the cotyledon, considering the sample weight measured.

Figure 6 represents a two-dimensional (2D) EPR image of pigments displaying radical distributions. Based on the line width and applied gradients, the EPRI in the samples was approximately 100 µm. Red indicates high radical concentrations and corresponds to the seed coats; indeed, radicals were spread over the entire seed coats. However, the EPR image obtained showed a non-uniform distribution, as the shape of the seed coat is not a smooth round surface. Figure 6B displays the EPR images at a 90-degree rotation of the sample in Figure 6A. Here, the red color corresponds to the edge of the seeds. The red-dotted arrow indicates the edge of the seeds in Figure 6. The 2D-EPR techniques provide further insight into the radical intermediate. Stable organic radicals in seeds can be produced during ROS and/or RNS scavenging [33]. ROS and RNS are involved in the regulation of various processes in plants, such as germination, flowering, and senescence [31,34]. ROS generated in plant seeds react with phenolic or flavonoid compounds and produce stable radicals in the seed [32]. The stable organic radicals are not reactive; thus, these stable radicals can be an indirect indicator of ROS. Dormancy and various stages of seeds have been previously examined to identify ROS-related reactions [11]. Significant radical intensity changes in apple seeds have been observed under cold stratification [32]. Thus, stable organic radicals may be an indicator of antioxidant-related activity in the seeds.

Figure 7 displays the 2D-EPRI of the cotyledon. In the right-hand panel, the dotted arrow indicates the approximate size of the seeds. The top portion of the cotyledon corresponds to the red region of the image, indicating that stable radicals were produced by the oxidation of compounds, such as fatty acids (e.g., oleic acid and stearic acid) in the cotyledon. The relative radical intensity was much lower than that of the seed coat. Notably, the top portion of the seed coat had a small hole.

## 4. Conclusions

In summary, for many years, we used the cotyledon of the *C. japonica* seeds without paying attention to the seed coats. In the present investigation, the hydrophilic and lipophilic extractions of *C. japonica* seed coats revealed various antioxidant phytochemicals as verified by HPLC chromatograms. The *C. japonica* seed coat extracts also exhibited antioxidant activity in both in vitro and cell-based studies. X-band EPR detected paramagnetic species in the *C. japonica* seeds. The EPR image displayed stable radicals located throughout the seed coat and also a lesser amount present in the cotyledon. Thus, the present study demonstrated that EPR is a useful and detailed method that revealed that *C. japonica* seed coats are a useful source of phytochemical antioxidants.

## Figures and Tables

**Figure 1 antioxidants-09-00493-f001:**
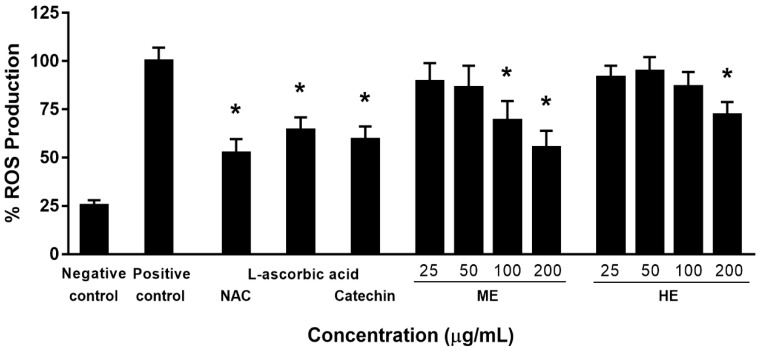
The effects of *C. japonica* seed coat extracts on H_2_O_2_-induced intracellular reactive oxygen species (ROS) production in peripheral blood mononuclear cells (PBMCs). Untreated cells were used as a negative control. *N*-acetylcysteine (NAC, 80 μM), L-ascorbic acid (250 µM), and catechin (125 µM) were used as positive controls. * Data represent the mean ± SD of three independent experiments. *p* < 0.05 versus positive control. ME: methanolic extract, and HE: hexane extract.

**Figure 2 antioxidants-09-00493-f002:**
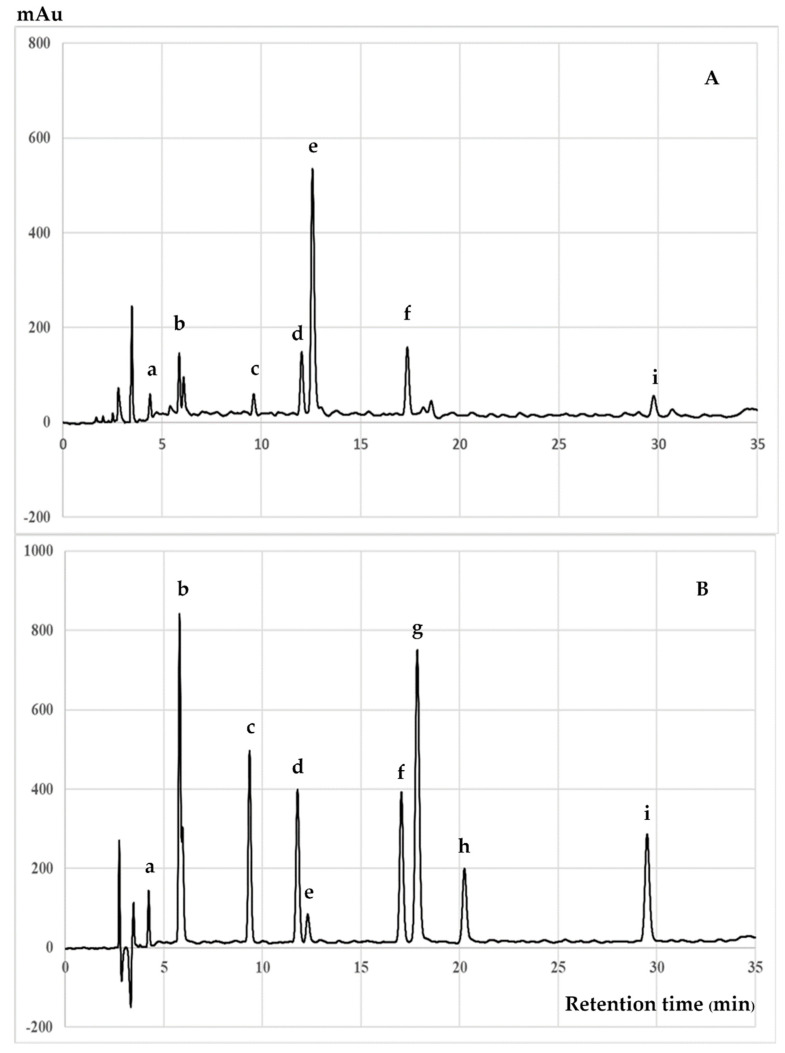
HPLC chromatograms of (**A**) the methanolic extract of *C. japonica* seed coats and (**B**) mixed catechin and related compounds were used as standards. The peaks are (**a**) gallic acid, (**b**) gallocatechin, (**c**) epigallocatechin, (**d**) caffeine, (**e**) catechin, (**f**) epicatechin, (**g**) epigallocatechin gallate, (**h**) gallocatechin gallate and (**i**) epicatechin gallate.

**Figure 3 antioxidants-09-00493-f003:**
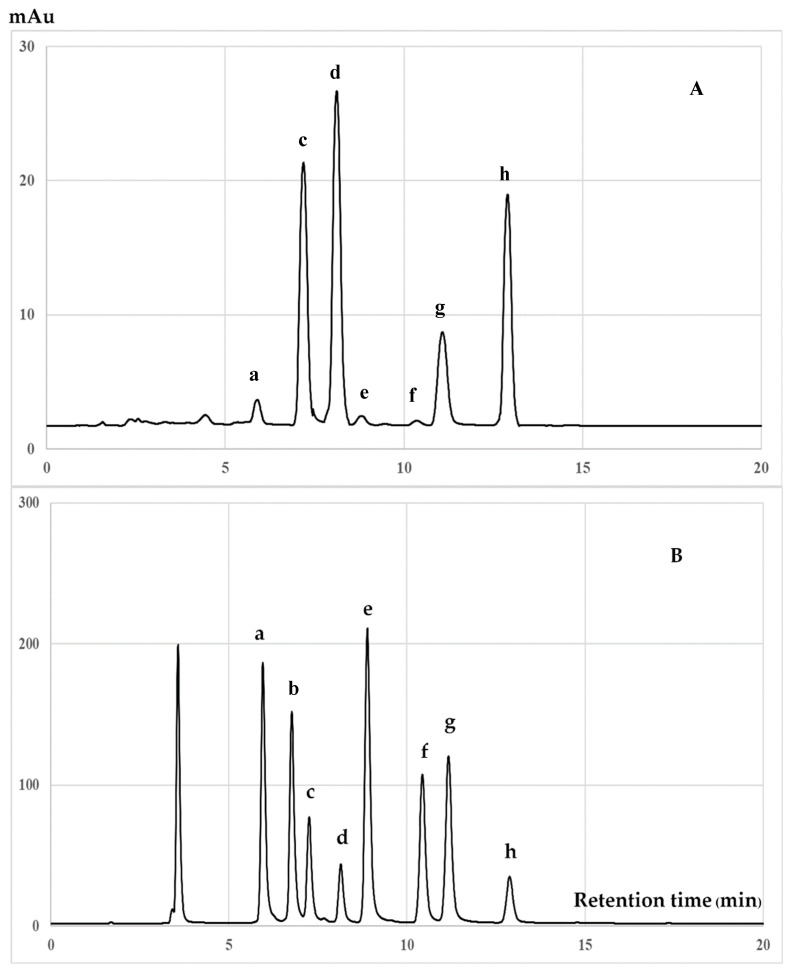
HPLC chromatograms of hexane extract of (**A**) hexane extract of *C. japonica* seed coats and (**B**) mixed tocotrienols and tocopherols standards. The peaks are (**a**): δ-tocotrienol; (**b**): β-tocotrienol; (**c**): γ-tocotrienol; (**d**): α-tocotrienol; (**e**): δ-tocopherol; (**f**): β-tocopherol; (**g**): γ-tocopherol; and (**h**): α-tocopherol.

**Figure 4 antioxidants-09-00493-f004:**
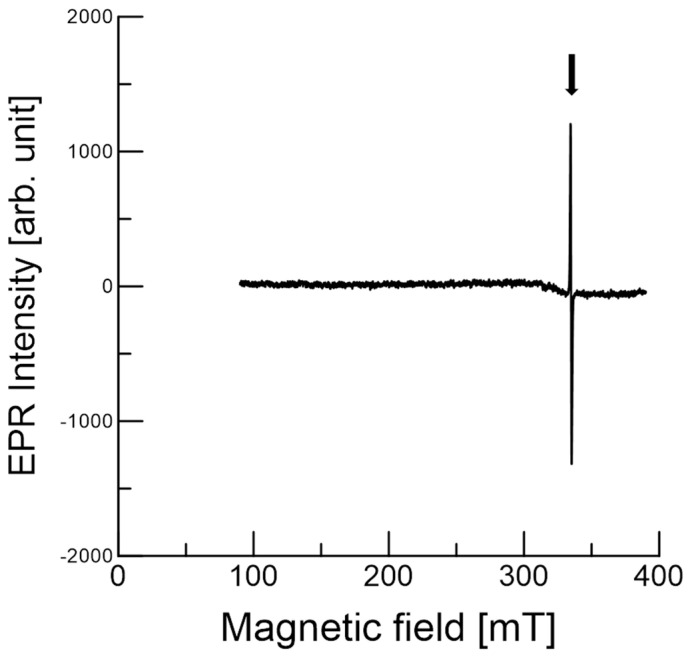
Electron paramagnetic resonance (EPR) spectra of a whole *C. japonica* seed at ambient temperature. The magnetic field center was 240.0 mT, with a 300 mT scan width. Each spectrum was obtained with a single scan. The arrow indicates the stable organic radicals.

**Figure 5 antioxidants-09-00493-f005:**
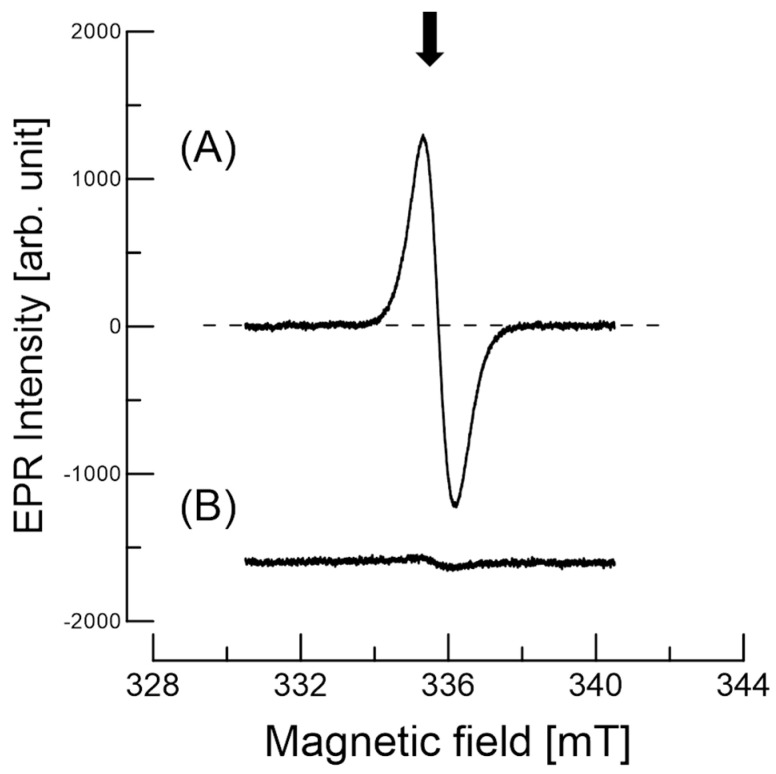
EPR spectra obtained for the g~2.002 region of (**A**) the *C. japonica* seed coats at ambient storage conditions and (**B**) the cotyledon of the seeds. Each spectrum was obtained with a single scan. The arrow indicates the stable radical center.

**Figure 6 antioxidants-09-00493-f006:**
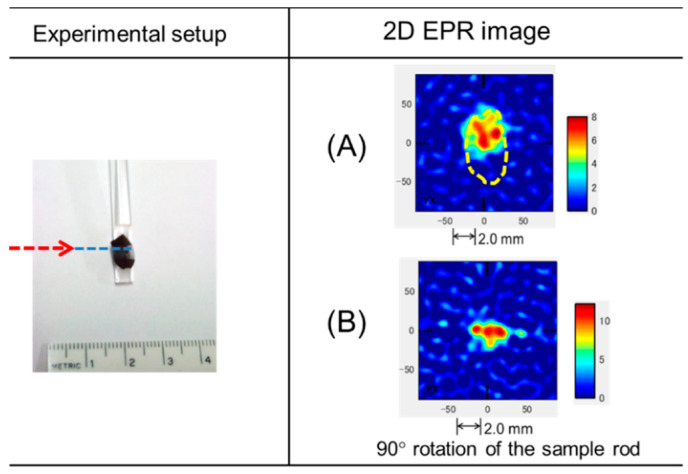
The left-hand panel shows the measurement set-up. The right-hand panel displays a 2D-EPR image of a *C. japonica* seed. The red-dotted arrow indicates the edge of the seeds. The dotted line in (**A**) shows the approximate size of the samples. The EPR image in (**B**) was taken at a 90-degree rotation compared to (**A**) position.

**Figure 7 antioxidants-09-00493-f007:**
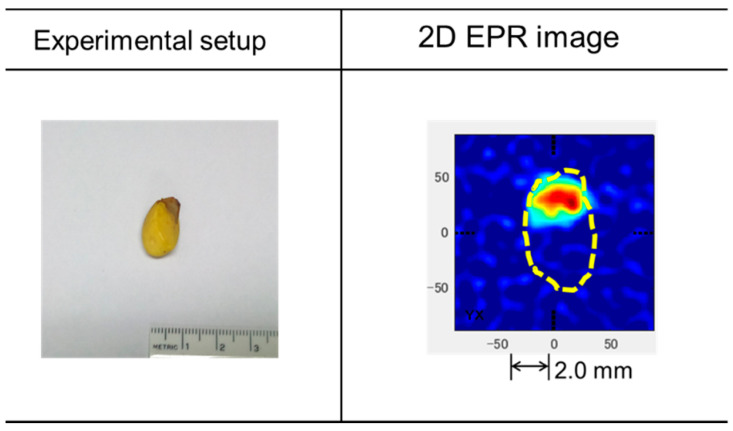
The left-hand panel displays the cotyledon. The right-hand panel displays a 2D EPR image of a *C. japonica* cotyledon. The dotted line indicates the approximate size of the cotyledon.

**Table 1 antioxidants-09-00493-t001:** The 50% inhibition concentration (IC_50_) of nitric oxide, superoxide anion scavenging activities, and the inhibitory effect on lipid peroxidation.

Samples/Positive Control		IC_50_ (µg/mL)	
	Nitric Oxide	Superoxide Anion	Lipid Peroxidation
ME	22.64 ± 1.07 ^c^	14.5 ± 0.82 ^c^	59.58 ± 2.81 ^d^
HE	88.84 ± 2.23 ^d^	74.5 ± 2.26 ^d^	20.98 ± 2.17 ^b^
Catechin	14.86 ± 0.84 ^b^	9.61 ± 0.31 ^b^	30.64 ± 1.32 ^c^
Curcumin	9.14 ± 0.68 ^a^	ND	ND
L-ascorbic acid	ND	6.91 ± 0.26 ^a^	ND
α-Tocopherol	ND	ND	14.32 ± 0.93 ^a^

ME: Methanolic extracts; HE: Hexane extracts. All values (^a–d^) are expressed as the mean ± standard deviation (*n* = 3). Different letters for each method indicate a significant difference (*p* < 0.05). ND: not determine.

**Table 2 antioxidants-09-00493-t002:** The IC_50_ of nitric oxide and iNOS production induced by combination with LPS-IFN-γ in RAW 264.7 cells.

Samples/Positive Control	IC_50_ (µg/mL)	
	Nitric Oxide	iNOS
ME	22.78 ± 1.79 ^c^	32.68 ± 1.16 ^C^
HE	33.95 ± 2.08 ^d^	39.63 ± 1.78 ^D^
Catechin	12.55 ± 0.77 ^b^	17.45 ± 1.29 ^B^
Curcumin	7.51 ± 0.69 ^a^	9.52 ± 0.63 ^A^

ME: Methanolic extracts; HE: Hexane extracts. All values (^a–d, A–D^) are expressed as mean ± standard deviation (*n* = 3). Different letters for each method indicate a significant difference (*p* < 0.05).

**Table 3 antioxidants-09-00493-t003:** The amount of phytochemical compounds in the ME and HE extracts. All values are expressed as mean ± standard deviation (*n* = 3). nd: not detected.

Compounds	Amount (mg/g Extract)		Compounds	Amount (mg/g Extract)	
	ME	HE		ME	HE
Gallic acid	0.64 ± 0.10	nd.	δ-Tocotrienol	nd.	1.44 ± 0.35
Gallocatechin	1.98 ± 0.22	nd.	β-Tocotrienol	nd.	nd.
Epigallocatechin	1.24 ± 0.11	nd.	γ-Tocotrienol	1.16 ± 0.28	15.7 ± 0.52
Caffeine	3.69 ± 0.27	0.86 ± 0.12	α-Tocotrienol	1.49 ± 0.23	18.2 ± 0.76
Catechin	9.46 ± 0.45	nd.	δ-Tocopherol	nd.	0.82 ± 0.29
Epicatechin	3.55 ± 0.20	nd.	β-Tocopherol	nd.	0.37 ± 0.14
Epigallocatechin gallate	0.52 ± 0.08	nd.	γ-Tocopherol	0.47 ± 0.15	6.27 ± 0.39
Epicatechin gallate	0.86 ± 0.13	nd.	α-Tocopherol	0.65 ± 0.19	13.6 ± 0.53

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
