# Peer review of "Detection of Antioxidant Phytochemicals Isolated from Camellia japonica Seeds Using HPLC and EPR Imaging"

_antioxidants, 2020, doi:10.3390/antiox9060493_

Round 1

Reviewer 1 Report

The article is quite difficult to read, in part probably due to language/English difficulties. I recommend the authors to perform extensive editing of English language.

The introduction is also confusing, eg authors mixture the formation and antioxidants reactions in the seed coats without appropriate meaning clarification. Moreover, what is the meaning of "pigments in the seed coat have not been investigated to date, in line 62"?

It is also necessary to point clearly what has been reported before for this plant (leaves and other parts including seed coat) and highlight the novelty of this work. 

References in the text are not cited according to journal rules. Please revise. Also, please reduce references in the methods. 

The identification of phenolic compounds was solely based on the comparison of retention time with standards? if yes, how can they be sure about the type of compounds? based on other works? ... at least UV spectral data should be considered. 

The results must be clearly presented. Eg note that they consider the same terminology for nitric oxide (in chemico vs cell assay). Looking at Table 1 vs Table 2, readers wont be able to distinguish. So, Tables must be more informative. 

Fig 1- 250 mM for Ascorbic acid and 125 mM for catechin?! These are huge values: are they correct?

Reviewer 2 Report

The present study investigated the in vitro and in cell antioxidant activity, the formation of stable radicals and compounds by HPLC and EPR imaging in Camellia japonica seeds. This is a descriptive study which shows the antioxidant activity of two extracts (ME and HE) from Camellia japonica seeds. The authors, evaluated the seed antioxidant capacity to stabilize ROS and RNS in vitro and modulate the activity of iNOs and ROS levels in cells. Furthermore, the authors indicate the presence of hydrophilic antioxidant compounds in ME extract and the presence of lipophilic antioxidant compounds in HE compounds, which is to be expected from the nature of the extracts. The more interesting is the study by EPR imaging where evaluated the localization of the stable radicals in the seeds.

This paper has some important points have to be clarified:

  • In the introduction the authors, not indicate the hypothesis of the study
  • Material and methods: Methods not are clear. Is necessary to explain the methodology with more detail. For example, What is the concentrations of the reactive used to evaluate the scavenging effects of superoxide radical? And for lipid peroxidation? and how expressed the ROS production in cells?

Line 145-1448, the authors indicates that evaluated cellular DNA, protein concentration and cell viability, Why?.

The authors should include a section of statistical analyses

  • Results and discussion. The results are very descriptive and not are a clear discussion. The authors could discuss with more detail the novelty of the antioxidant activity and compounds in the different seeds extracts, their potential utility and what is the relationship with the EPR results.

In line 201-202 the authors said “Interestingly, HE of C. japonica seed exhibited a higher antioxidant activity through inhibition effect on lipid peroxidation than that of ME of C. japonica seed coat” Why is interestingly? in my opinion that is known by their lipophilic nature.

In table 2, the authors should indicate that the study of antioxidant capacity is in cells.

In figure 5, the authors indicated that control cell is untreated cells, but the cells were treated with H2O2 and what it means*? The authors should indicate the statistical used for the study.

In the chromatographic analysis of biochemical compounds by HPLC,  Why the authors not express the results quantitatively ?, the chromatographic profile not shows the levels of the compounds in the extracts for to obtained a conclude about of their contribution in the seed extract.

The Figure 5, is a simple scheme of the action of antioxidant, where is the stable intermediate (line 278) that postulate the authors?  The discussion could be enlarged to explain the differences between extracts and the differences obtained between seed coat and cotyledon by EPR spectra.

 The authors would revise the conclusion and not to resume the results obtained.

Reviewer 3 Report

The article deals with a comprehensive and innovative analysis of hydrophilic and lipophilic antioxidants compounds in the seeds of Camellia japonica.

A strong novelty is the observation and characterization of paramagnetic species by means of EPR (and EPRI), specifically stable organic radicals, as indicators of the scavenging of ROS and RNS.

The experimental design is well done, and the results are clearly presented.

Few language and style errors, and few unclear, unsupported or undocumented statements require a revision.

All my comments and suggestions are available in the herein enclosed document. The Authors are invited to consider all such comments, respond and react accordingly.

Round 2

Reviewer 1 Report

Albeit authors have considerably improved the manuscript, there is still the need clarify some issues.

Identification of phenolic compounds only based on RT comparison (even with standards) is not a reliable method. Please add UV spectral data from eluting peaks (confirming that they correspond those of standards).
Moreover, please clarify the eluting program used for phenolic compounds analysis. Are there any more peaks in the chromatogram eluting after 35 min?
Is the program for tocopherol/tocotrienols the same as described in ref 21?
“The ME of C. japonica seed coats contained hydrophilic antioxidant compounds, namely gallic acid, gallocatechin, epigallocatechin, caffeine, catechin, epicatechin, epigallocatechin gallate, and epicatechin gallate, which were not detected in the HE of C. japonica seed coats”. Have these compounds been previously described for C. japonica seed coats ? Or in seed coats from other Camellia species?

Reviewer 2 Report

The authors well addressed to the comments. This reviewer has no more question and query.

Round 3

Reviewer 1 Report

Thank you for the clarifications. I would like you to further clarify:
What is the reason for using an isocratic HPLC program? “… 10% acetonitrile in 0.1% acetic acid and de-ionized water was used for the mobile phase at the flow rate of 1.0 mL/min with the detection wavelength at 280 nm.” In such conditions only polar compounds will be eluted from the column but naturally, the fact that no more peaks were detected does not mean that there are no more phenolic compounds in the extracts …
